# Anomalous metallic behaviour in the doped spin liquid candidate $\kappa$-(ET)$_4$Hg$_{2.89}$Br$_8$

Hiroshi Oike [1,3], Yuji Suzuki [1], Hiromi Taniguchi[2], Yasuhide Seki[1], Kazuya Miyagawa[1] & Kazushi Kanoda [1]

Quantum spin liquids are exotic Mott insulators that carry extraordinary spin excitations. Therefore, when doped, they are expected to afford metallic states with unconventional magnetic excitations. Here, we report experimental results which are suggestive of a doped spin liquid with anomalous metallicity in a triangular-lattice organic conductor. The spin susceptibility is nearly perfectly scaled to that of a non-doped spin liquid insulator in spite of the metallic state. Furthermore, the charge transport that is confined in the layer at high temperatures becomes sharply deconfined on cooling, coinciding with the rapid growth of spin correlations or coherence as signified by a steep decrease in spin susceptibility. The present results substantiate the desired doped spin liquid and suggest a strange metal, in which the coherence of the underlying spin liquid promotes the deconfinement of charge from the layers while preserving the non-Fermi-liquid nature.

[1] Department of Applied Physics, University of Tokyo, Bunkyo-ku, Tokyo 113-8656, Japan. [2] Graduate School of Science and Engineering, Saitama University, Saitama 338-8570, Japan. [3] Present address: RIKEN Center for Emergent Matter Science (CEMS), Wako, Saitama 351-0198, Japan. Correspondence and requests for materials should be addressed to H.O. (email: hiroshi.oike@riken.jp) or to K.K. (email: kanoda@ap.t.u-tokyo.ac.jp)

Strongly interacting electrons show a variety of ground states and distinctive elementary excitations, depending on interaction strength, lattice geometry, and band filling. In half-filled systems, the strong on-site Coulomb interaction $U$ exceeding the bandwidth $W$ prevents electrons from doubly occupying a site, and drives the system into a Mott insulating state. Although Mott insulators typically exhibit magnetic order at low temperatures, spin frustration in a triangular-lattice system is theoretically expected to prevent magnetic ordering and to lead to an exotic spin state[1]. Indeed, several materials with quasi-triangular lattices have been found to host spin liquids without magnetic ordering[2, 3]. What phases emerge when carriers are injected into spin liquids by doping is an intriguing issue[4–9]. However, no doped spin liquids have been realized experimentally, although a doped spin liquid has been proposed as a model for high-$T_c$ superconductivity in copper oxides[4], which are not spin liquids before being doped.

A promising candidate for doped spin liquids is the organic conductor $\kappa$-(ET)$_4$Hg$_{2.89}$Br$_8$ (abbreviated as $\kappa$-HgBr)[10], which has conducting ET layers sandwiched between insulating layers composed of Hg and Br ions (Fig. 1). In the conducting layers, the ET dimers form a nearly isotropic triangular lattice with respect to transfer integrals between dimers as in the spin liquid insulator $\kappa$-(ET)$_2$Cu$_2$(CN)$_3$ (abbreviated as $\kappa$-Cu$_2$(CN)$_3$) (Fig. 1)[11]. The spin liquid nature of $\kappa$-Cu$_2$(CN)$_3$ has been intensively studied and is revealed to contain unexpected features such as field-induced inhomogeneity and the so-called 6K-anomalies, suggesting still unknown but fertile phenomena underlying in the spin liquid state[1, 12]. In the insulating layers, the Hg ions form a sub-lattice incommensurate with the Br and ET sub-lattice. This results in a fixed non-stoichiometric composition of 2.89[13], and the deficiency from 3.0 causes 11% hole doping in the half-filled band in the conducting layer. Thus, carrier doping is created in the triangular-lattice system. In actuality, the organic conductor $\kappa$-HgBr is a metal. The in-plane resistivity $\rho_{//}$ shows non-Fermi liquid (non-FL) behavior at low pressures, and it exhibits Fermi liquid (FL) behavior at high pressures above 0.5 GPa[14, 15], where an increase in carrier density is indicated by the Hall coefficient, in spite of the fixed band filling[15]. Thus, the $\kappa$-HgBr is a doped Mott insulator with strongly prohibited double occupancy (or

strongly bound doublon–holon pairs) at low pressures or a correlated metal with all the electrons contributing to carriers at high pressures, analogous to the pressure-driven Mott metal-insulator transition at half filling[15].

In this article, we describe the results of spin susceptibility and resistivity measurements, which probe the spin and charge sectors of the doped triangular-lattice organic conductor $\kappa$-HgBr. The spin susceptibility in the doped Mott insulator regime is nearly perfectly scaled to that of a non-doped spin liquid insulator, indicating that the spin sector of $\kappa$-HgBr behaves like spin liquid in spite of the metallic state. The non-Femi liquid nature of the charge sector is confirmed by the temperature-linear resistivity. Furthermore, a sharp non-metal-to-metal crossover in the out-of-plane resistivity coincides with a drop-off in spin susceptibility, indicating entanglement between spin correlations and interlayer charge coherence. The present results suggest the realization of a doped spin liquid that hosts a non-Fermi liquid in a triangular-lattice organic conductor.

## Results

**Spin susceptibility**. To assess the spin liquid nature of $\kappa$-HgBr in the doped Mott insulator regime, we measured the spin susceptibility $\chi_{\mathrm{spin}}$ (Fig. 2a). It is characterized by a linear temperature dependence at high temperatures and a rounded peak at 30–40 K, followed by a sharp decrease, which is consistent with earlier reports[16, 17]. The behavior of the $\chi_{\mathrm{spin}}$ differs considerably from that of isostructural metallic compounds with half-filled bands, $\kappa$-(ET)$_2$X (X=Cu(NCS)$_2$ and Cu[N(CN)$_2$]Br), which reside close to the antiferromagnetic ground state. They show only weakly temperature-dependent Pauli-paramagnetic susceptibility on the order of $4 \times 10^{-4}$ emu mol$^{-1}$ at most (Fig. 2a)[18], which is less than half of the peak value ($9 \times 10^{-4}$ emu mol$^{-1}$) found in $\kappa$-HgBr. Instead, the $\chi_{\mathrm{spin}}$ of the $\kappa$-HgBr is well reproduced by a series expansion of the triangular-lattice Heisenberg model, which is applicable down to $T \sim 0.2 J$ or lower[19] with a nearest-neighbor exchange interaction, $J$, of 140 K (Fig. 2a). The $\chi_{\mathrm{spin}}$ of the quantum spin-liquid Mott-insulator $\kappa$-Cu$_2$(CN)$_3$ is also known to comply with that of the triangular-lattice Heisenberg model with a $J$ of 250 K[1]. The differences in the $J$ values are reasonably explained by the differences in $U$ and

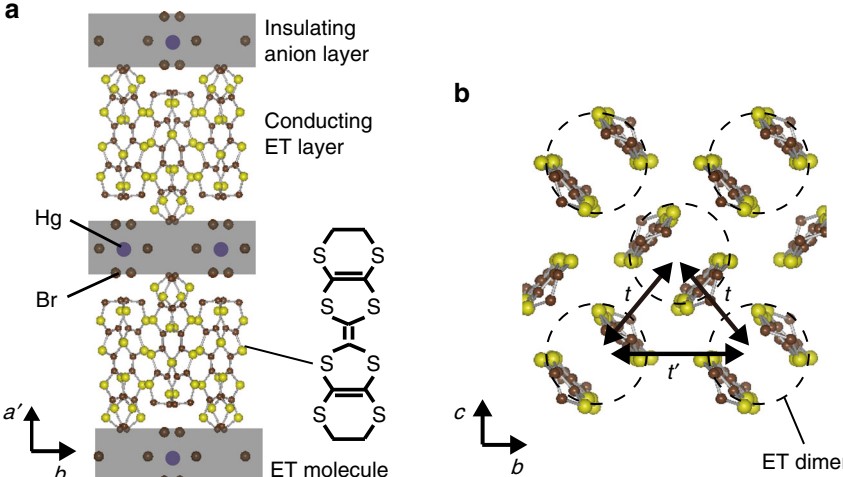

**Fig. 1** Crystal structure of $\kappa$-(ET)$_4$Hg$_{2.89}$Br$_8$ (denoted by $\kappa$-HgBr). **a** ET denotes bis(ethylenedithio)-tetrathiafulvalene. **b** Shows the conducting ET layer, which is parallel to the $b$-$c$ plane. The electronic bands of $\kappa$-ET compounds are well described by the tight binding of antibonding molecular orbitals in the ET dimers (indicated by broken circles). The lattice of dimers is modeled as a nearly isotropic triangular lattice with a $t'/t$ value close to unity, where $t$ and $t'$ are the transfer integrals between the antibonding orbitals of dimers forming a conduction band[11]. The non-stoichiometric composition of Hg, 2.89, originates from the incommensurability of the Hg sublattice with the Br and ET sublattices, and is precisely known by the incommensurability determined by X-ray diffraction[13]

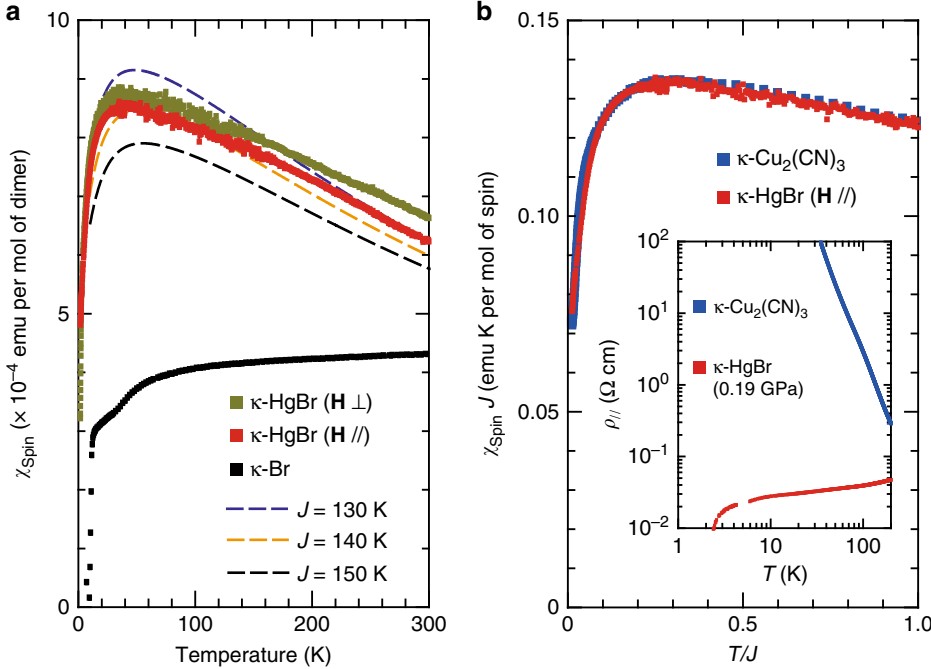

**Fig. 2** Evidence for a doped spin liquid. **a** Temperature dependences of the spin susceptibility $\chi_{spin}$ of $\kappa$-HgBr and $\kappa$-(ET)$_2$Cu[N(CN)$_2$]Br (denoted by $\kappa$-Br)[18]. The susceptibility under magnetic fields parallel and perpendicular to the conducting layers is indicated by red and green points, respectively. The broken lines represent the numerical curves obtained by the series expansion of the triangular-lattice Heisenberg model with a Padé approximant of order [7/7][19]. Comparing the calculations and the experimental data yields a $J$ value of 130–150 K. **b** Scaled $\chi_{spin}$ of $\kappa$-HgBr and $\kappa$-(ET)$_2$Cu$_2$(CN)$_3$ (denoted by $\kappa$-Cu$_2$(CN)$_3$)[2]. $J\chi_{spin}$ is plotted against $T/J$, where $\chi_{spin}$ is defined per mole of spin and $J$ is in unit of $k_B$. The inset shows the in-plane resistivity of the two compounds

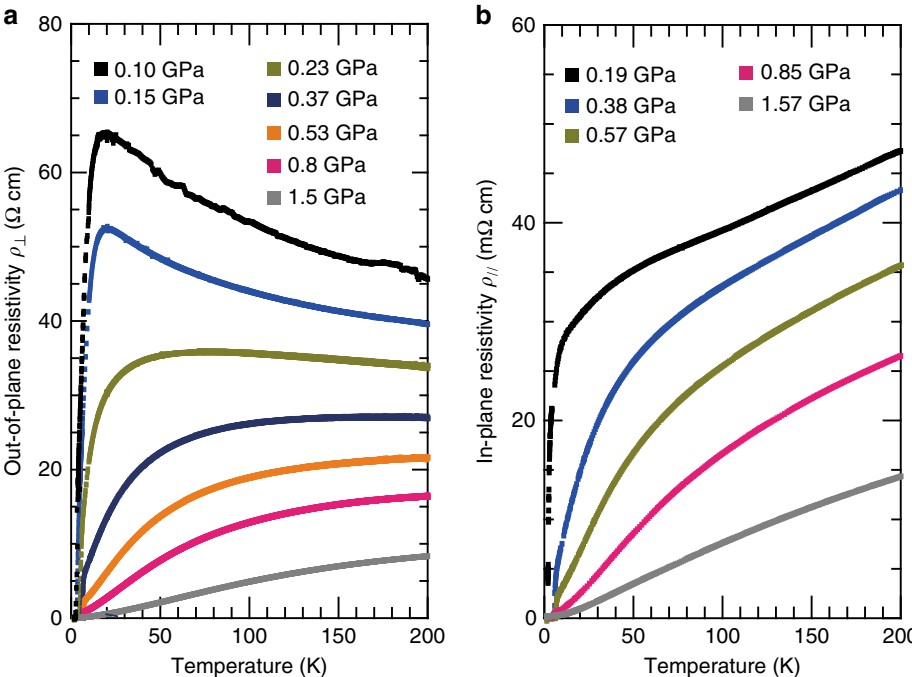

**Fig. 3** Out-of-plane and in-plane resistivities of $\kappa$-(ET)$_4$Hg$_{2.89}$Br$_8$. **a, b** Temperature dependences of out-of-plane resistivity $\rho_\perp$ (**a**) and in-plane resistivity $\rho_{//}$ (**b**) at several pressures. At ambient pressure, this material shows a number of spurious jumps in resistivity with temperature variation, very probably due to micro-cracking in the crystal, as encountered in several organic materials. Thus, applying finite pressures was necessary for obtaining reliable data

$t$ between $\kappa$-HgBr and $\kappa$-Cu$_2$(CN)$_3$ (see Methods). To compare the $\chi_{spin}$ of the two systems, we examined the scaling of $J\chi_{spin}$ against $T/J$[20]. Remarkably, the two results are indistinguishable, and it is noted that there is no adjustable parameter

(Fig. 2b). Only a slight difference in the temperature range below 0.1 $J$ is recognizable (Supplementary Fig. 1). The nearly perfect coincidence clearly shows that in the spin sector, $\kappa$-HgBr behaves nearly exactly like the spin liquid insulator, $\kappa$-Cu$_2$(CN)$_3$, with the

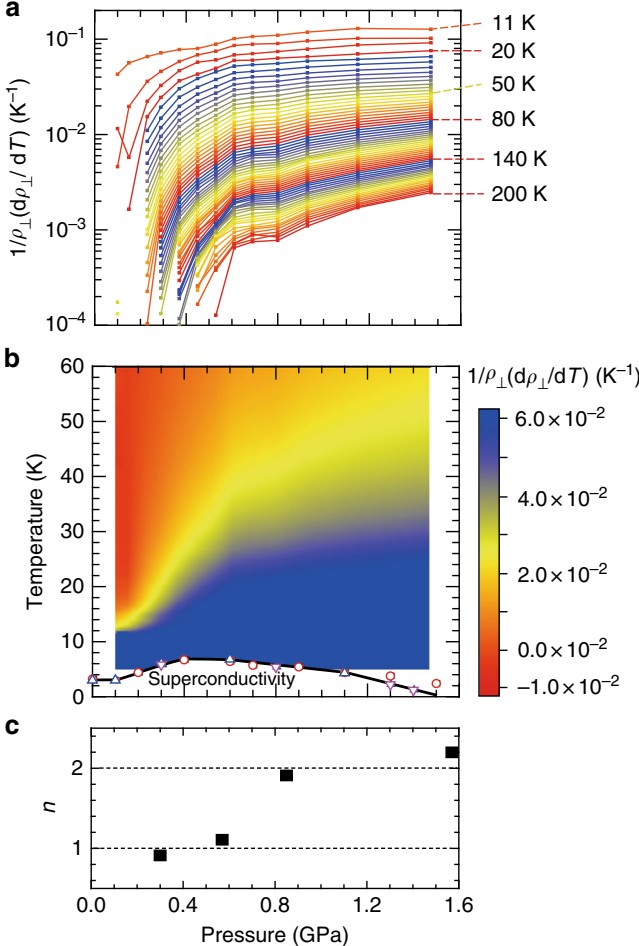

**Fig. 4** Pressure dependence of charge excitations. (**a**) Pressure dependence of $1/\rho_\perp(\mathrm{d}\rho_\perp/\mathrm{d}T)$. (**b**) Contour plot of $1/\rho_\perp(\mathrm{d}\rho_\perp/\mathrm{d}T)$ in the pressure–temperature plane. The solid line represents the superconducting transition temperatures determined by ac susceptibility measurements[15]. **c** Pressure dependence of power law behavior of $\rho_{//}$. The power $n$ is determined by fitting the form of $\rho_{//} = \rho_o + AT^n$ to the data of $\rho_{//}$ in a temperature range between superconducting transition temperature and 15 K (see Methods and Supplementary Fig. 3)

exception of the difference in $J$, while they behave quite differently in the charge sector (inset of Fig. 2b). This is a clear indication of spin–charge separation. A prominent feature in the scaled susceptibility is a sharp decrease at low temperatures, for example, below 0.1 $J$, which is beyond the temperature range of the series expansion. Considering that there is no magnetic ordering in these systems[2, 21], the sharp decrease in the susceptibility is not a simple indication of the growth of the 120° spin correlation, which would end up with a long-range order in the Heisenberg model. Instead, the sharp decrease signifies the increasing correlation or coherence inherent in the spin liquid. Interestingly, NMR relaxation rate shows an accelerated decrease with temperature around 10 K in both systems[2, 17], which may indicate the crossover from the 120° short range order to quantum spin liquid. Thus, the present result provides the first direct indication of the spin liquid nature of a metallic phase revealed by magnetic measurements.

**Out-of-plane resistivity**. Having justified that $\kappa$-HgBr hosts a doped spin liquid, we proceeded to further characterize the

non-FL state using out-of-plane transport, which probes the nature of the conducting state through the inter-layer tunneling of charge carriers to address the issue of charge excitations in the metallic state[22]. Figure 3a shows the $\rho_\perp$ of the $\kappa$-HgBr with the tuning pressure across the non-FL-to-FL transition/crossover point. At 0.1 and 0.15 GPa, the temperature dependence of the out-of-plane resistivity $\rho_\perp$ is non-metallic in a wide temperature range, where $\rho_{//}$ behaves like a metal (Fig. 3b). A peak is reached in the $\rho_\perp$ at ~20 K, followed by a steep decrease. Above 0.2 GPa, the peak in $\rho_\perp$ with respect to temperature is broadened and shifted towards higher temperatures. At pressures above 0.5 GPa, the temperature dependence of $\rho_\perp$ is metallic over a wide temperature range.

To characterize the out-of-plane metallicity, we plotted the rate of change in $\rho_\perp$ with respect to temperature, $1/\rho_\perp(\mathrm{d}\rho_\perp/\mathrm{d}T)$. As the pressure decreases, the interlayer metallicity begins to decrease at ~0.5 GPa for all temperatures except below 20 K (Fig. 4a). The contour plot of $1/\rho_\perp(\mathrm{d}\rho_\perp/\mathrm{d}T)$ highlights how the interlayer-metallic region extends to higher temperatures for higher pressures (Fig. 4b). There are three distinctive regions; $P < 0.3\,\mathrm{GPa}$, $0.3 < P < 0.6\,\mathrm{GPa}$ and $P > 0.6\,\mathrm{GPa}$. The high-pressure region ($P > 0.6\,\mathrm{GPa}$) corresponds to the FL region identified by the behavior of $\rho_{//}$, which exhibits quadratic temperature dependence roughly in the blue-colored region (Fig. 4c)[15]. In $P < 0.3\,\mathrm{GPa}$, the peculiar nonmetal-to-metal crossover sharply occurs only in the out-of-plane direction at low temperatures, 10–15 K (Fig. 3a) in the non-FL regime, where temperature-linear dependence of $\rho_{//}$ is observed (Fig. 4c)[14, 15]. The intermediate region ($0.3 < P < 0.6\,\mathrm{GPa}$) is a transition region between the two regimes.

To examine the possible involvement of superconducting fluctuations in the nonmetal-to-metal crossover in $P < 0.3\,\mathrm{GPa}$, we measured $\rho_{//}$ and $\rho_\perp$ under magnetic fields perpendicular to the layers (Fig. 5). Above the superconducting transition temperature $T_c$, both $\rho_{//}$ and $\rho_\perp$ are not affected by the magnetic fields that are large enough to extinguish the superconductivity. This rules out the role of superconductivity in the low-temperature restoration of the interlayer coupling. It is remarkable that the non-FL nature persists down to the lowest temperature studied, 1.8 K, when the superconductivity disappears. This observation is contrary to the FL behaviors of non-doped systems in the vicinity of the Mott transition[23].

## Discussion

A clue to the puzzling interlayer transport at low pressures is found in the steep decrease of $\chi_{\mathrm{spin}}$, which occurs in the same temperature range (below 0.1 $J$) as the nonmetal-to-metal crossover in the $\rho_\perp$, indicating the recoupling of charge and spin. Fermionic spinon excitations have been proposed as a model of spin liquids[24], which have finite $\chi_{\mathrm{spin}}$ values at low-temperatures as in the present system (Supplementary Fig. 1). In this context, the coincidence of the transport and magnetic anomalies suggests that the interlayer transport of doped carriers is enhanced by the formation of coherent spinons. It is noted, however, that the spin-charge coupling is weak because the non-Fermi-liquid nature persists even after the nonmetal-to-metal crossover in $\rho_\perp$ occurs. If the spinons and doped carriers were strongly coupled, conventional quasi-particles would result, leading to a Fermi liquid with large Fermi surfaces at low temperatures. The absence of large Fermi surfaces is supported by the Hall coefficient, which indicates a considerable decrease in the carrier density at low pressures[15]. Thus, we suggest that an extraordinary quantum fluid in which spin and charge are weakly coupled emerges in a doped spin liquid, whereas spin and charges are separated in the high temperature range. A slight difference

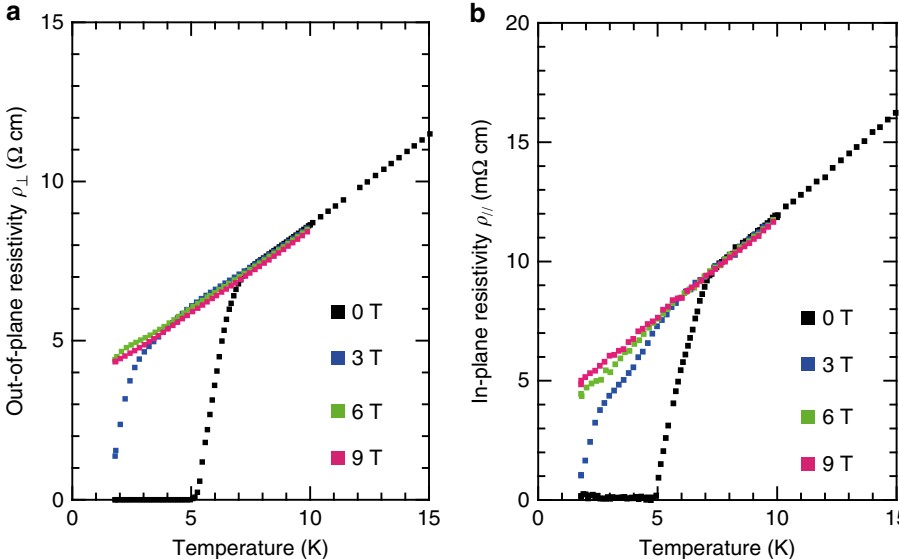

**Fig. 5** Magnetic field dependence of resistivity. **a**, **b** Out-of-plane resistivity $\rho_\perp$ (**a**) and in-plane resistivity $\rho_{//}$ (**b**) under magnetic fields of 0, 3, 6 and 9T applied perpendicular to the conducting plane at a pressure of 0.3 GPa

between the susceptibilities of the undoped and doped systems at temperatures below 0.1 $J$ (Supplementary Fig. 1) could be a signature of the weak spin–charge coupling.

The peculiarity of the doped spin liquid also appears in the electronic specific heat coefficient, $\gamma = 55$ mJ mol$^{-1}$ K$^{-2}$, which is extraordinarily large among organic conductors; e.g., $\gamma = 25$ and 22 mJ mol$^{-1}$ K$^{-2}$ for the half-filled metals, $\kappa$-(ET)$_2$X with X=Cu (NCS)$_2$ and Cu[N(CN)$_2$]Br, respectively[25]. The Wilson ratio given by $(\pi^2/3)(\chi_{\rm spin}/\mu_B^2)/(\gamma/k_B^2)$ for $\kappa$-HgBr is 0.63, an anomalous value that considerably deviates from the values for the half-filled systems, 0.95 and 0.98, respectively, typical values for Fermi liquids[26, 27]. On the other hand, in the spin liquid system $\kappa$-Cu$_2$(CN)$_3$, the $\gamma$ value of 13 mJ mol$^{-1}$ K$^{-2}$ is discussed as a hallmark of the spinon Fermi surfaces in conjunction with the acceptable Wilson ratio, 1.6, as fermionic excitations. Notably, another spin liquid insulator, Me$_3$EtSb[Pd(dmit)$_2$]$_2$, also shows nearly the same Wilson ratio. Assuming this value for the spin sector in $\kappa$-HgBr, $\gamma$ becomes 22 mJ mol$^{-1}$ K$^{-2}$, which explains only a fraction of the observed $\gamma$ value. Given that spin and charge are only weakly coupled with each other, we ascribe the remaining large value, 33 mJ mol$^{-1}$ K$^{-2}$, to the charge channel opened by 11% doped holes with the non-Fermi liquid nature.

An anomalous metal with a preserved spin–liquid nature, as observed here is theoretically suggested for multi-band systems with conduction electrons and localized spins on geometrically frustrated lattices[28–31] such as the Shastry–Sutherland lattice[32, 33] and pyrochlore lattices[34]. In the weak Kondo coupling regime, electrons in the conduction band become only weakly hybridized with the frustrated spins in a spin liquid state[31]. The predictions include the following similarities to the present observations in the single-band $\kappa$-HgBr system. First is that the anomalous metal emerges upon cooling from a situation in which the conducting and magnetic channels appear disentangled at high temperatures. Second is that the anomalous metal turns into a conventional FL, when the transfer integral in our system or the Kondo coupling in the multiband model is increased[35]. In this scenario, the non-FL-to-FL crossover at 0.5 GPa can be an indication of the strong spin–charge coupling leading to quasiparticle formation. Addressing the present issue may require reconciling the predictions for the multiband model with the present observations in the single-band system.

Seeking qualitatively different metallic phases from the Fermi liquid is a fundamental challenge in condensed matter physics. The present results indicate that a doped spin–liquid is realized in a real material, and it is a non-Fermi liquid in the charge sector and a spin–liquid in the spin sector, where the two sectors are weakly coupled to each other. Notably, superconductivity emerges in this system. What kind of Cooper pairing occurs in this strange metal is an issue of future work.

## Methods

**Band calculations**. In the present study, we used the band parameters based on the extended Hückel method and tight-binding approximations with the dimer model. The lattice parameters used in the calculations are reported in refs. [36, 37]. The $t'/t$ values[38–41] and the predicted ground states[42–49] are known to depend on the theoretical methods employed; however, the fact that the same (quantum chemical) method gives nearly the same $t'/t$ values, 1.02 for $\kappa$-HgBr and 1.06 for $\kappa$-Cu$_2$(CN)$_3$, suggests the similar degrees of spin frustration in the two materials. The difference in $J$ between $\kappa$-HgBr and $\kappa$-Cu$_2$(CN)$_3$ is reasonably explained by differences in $U$ and $t$, where $U$ is defined for a dimer of BEDT-TTF molecules and is estimated by $2t_{\rm dimer}$ with $t_{\rm dimer}$ the intra-dimer transfer integral in the limit of strong correlation on a ET molecule. In case that $U$ is sufficiently larger than $t$, $J$ is approximately given by $4t^2/U$, which yields 18.4 meV (210 K) for $\kappa$-HgBr and 26.8 meV (310 K) for $\kappa$-Cu$_2$(CN)$_3$ according to the band–structure calculations. These estimates qualitatively explain the magnitudes and material dependence of the $J$ values obtained by fitting the Heisenberg model to the experimental data; $J = 140$ K for $\kappa$-HgBr and 250 K for $\kappa$-Cu$_2$(CN)$_3$.

**Crystal growth and transport and susceptibility measurements**. The single crystals of $\kappa$-HgBr and $\kappa$-Cu$_2$(CN)$_3$ used in the present study were grown by the conventional electrochemical methods. The magnetic susceptibility was measured using a SQUID magnetometer (Quantum Design MPMS XL-7). The spin susceptibility was obtained by subtracting the core-diamagnetic contribution from the measured susceptibility. We confirmed that $\chi_{\rm spin}$ is nearly sample-independent in the normal state above 3 K (Supplementary Fig. 2). The in-plane and out-of-plane resistivities were measured by the conventional four-probe method using a source meter (Keithley 2400) and a nanovolt meter (Keithley 2182A). Pressure was applied with the use of a dual structured clamp-type cell formed by BeCu and NiCrAl cylinders. Daphne7373 oil, which does not solidify up to 2.2 GPa at room temperature, was used as a pressure-transmitting medium. The values of pressure reported here is the internal pressure at low temperatures, which is lower than the external pressure at room temperature due to solidification of the pressure medium and friction in the pressure cell.

**Power-law analysis of in-plane resistivity**. To characterise the power law behavior of $\rho_{//}$ at low temperatures, the resistivity data presented in Fig. 3b are fitted by the form of $\rho_{//} = \rho_{\rm o} + AT^n$ in a temperature range between the super-conducting transition temperature and 15 K. As shown in Supplementary Fig. 3,

the fitting works well for 0.57, 0.85, and 1.57 GPa, and the pressure dependence of power $n$ is consistent with the previous studies[14, 15] in that the value of $n$ changes from 1 to 2 under pressure. Concerning the data fitting for 0.19 and 0.38 GPa, $\rho_o$ takes negative values, which are unphysical, and thus the $n$ values deduced are considered not to be reliable. To obtain reliable $n$ value in this low pressure range, we need resistivity data at lower temperatures with superconductivity suppressed by magnetic field. Figure 5b shows resistivity data at 0.3 GPa under a magnetic field of 9 T, which allow a physically valid power-law fitting, as shown in Supplementary Fig. 3f.

**Data availability**. The data that support the findings of this study are available from the corresponding authors on request.

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

## Acknowledgements

We thank T. Koretsune, T. Senthil, N. Nagaosa, M. Ogata, and J. Ibuka for their useful comments. This work was supported in part by JSPS KAKENHI under Grant Nos.

20110002, 25220709, 24654101, and 11J09324 and by the US National Science Foundation under Grant No. PHYS-1066293 and the hospitality of the Aspen Center for Physics.

## Author contributions

Yu.S., H.T., and K.M. grew the single crystals used for the study. Yu.S., H.T., and K.M. conducted susceptibility measurements. H.O., Yu.S., H.T., and Ya.S. conducted transport experiments. H.O. analyzed the data. K.K. planned and supervised the project. H.O. and K.K. wrote the letter. All authors discussed the results and commented on the manuscript.

## Additional information

**Competing interests:** The authors declare no competing financial interests.

