## [Peer Review File · Nature Communications]

Reviewers' comments:

Reviewer #1 (Remarks to the Author):

The paper is a somewhat modified version of the manuscript previously submitted to [Redacted]. Besides some rewording, which I consider appropriate, noticeable changes have been made at three points: (i) In the revised introduction, the authors describe, in a more differentiated way, the present status of spin liquid research by pointing out that there are still some unexpected features. (ii) In the paragraph, where the spin susceptibility is discussed, the authors have included a statement on the NMR relaxation rate (Refs. 1, 17) indicating that there is also an accelerated decrease in the relaxation rate around 10 K for both systems $k\text{-Cu}_2(\text{CN})_3$ as well as $k\text{-HgBr}$. I find this information helpful, especially with respect to the discussion that follows in the subsequent paragraph where a link is made to the rather abrupt change in the out-of-plane resistivity. (iii) In the paragraph where the anomalous behavior of the interlayer transport is discussed, the authors added a sentence where they indicate a scenario to rationalize the small difference between the susceptibilities of the doped and undoped system below 0.1 J.

With these changes, the paper is more balanced in the way of reasoning and the conclusions that can be drawn. Also most of the exaggerated statements of what has been actually proven have been weakened or removed, except one statement in the introduction. I would strongly recommend to replace "...we provide experimental evidence for..." by a more appropriate statement such as "... we report experimental results which are suggestive of a doped ..."

In its revised form, I think the paper describes, in an appropriate way, an interesting experimental observation which will stimulate, further efforts in the important area of spin liquid research. Therefore, I recommend publication of the paper in Nature Communications.

Reviewer #3 (Remarks to the Author):

The authors report electrical transport and magnetic susceptibility measurements in $\kappa\text{-(ET)}_4\text{Hg}_{2.89}\text{Br}_8$ which is a non-stoichiometric organic compound displaying a triangular lattice.

Remarkably they find that the spin-susceptibility display the nearly same temperature dependence as the spin liquid insulator $\kappa\text{-(ET)}_2\text{Cu}_2(\text{CN})_3$, a compound extensively studied by Kanoda's group.

The first compound is metallic, albeit a strange metal, while the second is a Mott insulator characterized by an anomalous spin response. The authors claim that the first compound is the equivalent of a doped Mott insulator characterized by nearly non-interacting spin and charge channels. Pressure would increase their degree of interaction and eventually lead to conventional, Fermi liquid like behavior at low temperatures.

This is quite an intriguing study in a very relevant subject; the connection between spin-liquids, Mott insulating states and superconductivity. I do agree with authors that their study is relevant and of interest to a large community of researchers, in particular theoretical physicists. Therefore my recommendation is to publish this manuscript in Nature Communications.

However, it is a pity that the authors cannot measure transport under the exact same condition as the susceptibility, i.e. in absence of pressure, due to the emergence of micro-cracks upon cooling.

I am puzzled by the rather small and anomalous Wilson ratio. I would have expected a spin liquid to be near magnetic order and to display a relatively large Wilson ratio.

My only criticism is the method chosen to display the non-Fermi liquid to Fermi liquid crossover via a contour plot, where the authors display the rate of change of the inter-planar resistivity. Why don't they mention, for example, the exponent of the planar resistivity in the non-Fermi liquid region? It seems to be linear in temperature. Even better, can they also display the exponent of the planar resistivity, as extracted from the derivative of $\log(\rho - \rho_0) = n \log(T)$? Yes I agree that this method yields n values that rely heavily on the actual value of ρ_0 . But there are ways of circumventing this. The value of n is directly correlated with the Fermi liquid and the non-Fermi liquid behavior(s).

1. Regarding the Reviewer #1's comment 1-1, we replaced the sentence in the introduction "Here, we provide experimental evidence for a doped spin liquid..." by "Here, we report experimental results which are suggestive of a doped spin liquid...."
2. Regarding the Reviewer #3's comment 3-2, we added Fig. 3c, Supplementary Fig 3, and the new section "Power-law analysis of in-plane resistivity" in the Supplementary Information.

Reply to the Reviewer #1's comments

We thank the Reviewer #1 for illuminating advices that make the manuscript more balanced.

Reviewer #1 (Remarks to the Author):

The paper is a somewhat modified version of the manuscript previously submitted to [Redacted]. Besides some rewording, which I consider appropriate, noticeable changes have been made at three points: (i) In the revised introduction, the authors describe, in a more differentiated way, the present status of spin liquid research by pointing out that there are still some unexpected features. (ii) In the paragraph, where the spin susceptibility is discussed, the authors have included a statement on the NMR relaxation rate (Refs. 1, 17) indicating that there is also an accelerated decrease in the relaxation rate around 10 K for both systems $k\text{-Cu}_2(\text{CN})_3$ as well as $k\text{-HgBr}$. I find this information helpful, especially with respect to the discussion that follows in the subsequent paragraph where a link is made to the rather abrupt change in the out-of-plane resistivity. (iii) In the paragraph where the anomalous behavior of the interlayer transport is discussed, the authors added a sentence where they indicate a scenario to rationalize the small difference between the susceptibilities of the doped and undoped system below 0.1 J.

Comment 1-1:

With these changes, the paper is more balanced in the way of reasoning and the conclusions that can be drawn. Also most of the exaggerated statements of what has been actually proven have been weakened or removed, except one statement in the introduction. I would strongly recommend to replace "...we provide experimental

evidence for...“ by a more appropriate statement such as “... we report experimental results which are suggestive of a doped ...”

Reply 1-1:

Following the suggestion, we replaced the sentence in the introduction “Here, we provide experimental evidence for a doped spin liquid...” by “Here, we report experimental results which are suggestive of a doped spin liquid....”

Reply to the Reviewer #3’s comments

We thank the Reviewer #3 for reviewing our manuscript and indicating a point for improving the manuscript.

Reviewer #3 (Remarks to the Author):

The authors report electrical transport and magnetic susceptibility measurements in $\kappa\text{-(ET)}_4\text{Hg}_{2.89}\text{Br}_8$ which is a non-stoichiometric organic compound displaying a triangular lattice.

Remarkably they find that the spin-susceptibility display the nearly same temperature dependence as the spin liquid insulator $\kappa\text{-(ET)}_2\text{Cu}_2(\text{CN})_3$, a compound extensively studied by Kanoda's group.

The first compound is metallic, albeit a strange metal, while the second is a Mott insulator characterized by an anomalous spin response. The authors claim that the first compound is the equivalent of a doped Mott insulator characterized by nearly non-interacting spin and charge channels. Pressure would increase their degree of interaction and eventually lead to conventional, Fermi liquid like behavior at low temperatures.

This is quite an intriguing study in a very relevant subject; the connection between spin-liquids, Mott insulating states and superconductivity. I do agree with authors that the their study is relevant and of interest to a large community of researchers, in particular theoretical physicists.

Therefore my recommendation is to publish this manuscript in Nature Communications. However, it is a pity that the authors cannot measure transport under the exact same condition as the susceptibility, i.e. in absence of pressure, due to the emergence of micro-cracks upon cooling.

Comment 3-1:

I am puzzled by the rather small and anomalous Wilson ratio. I would have expected a spin liquid to be near magnetic order and to display a relatively large Wilson ratio.

Reply 3-1:

We also think that the value is really puzzling. The undoped spin liquids in $\kappa\text{-Cu}_2(\text{CN})_3$ and $\text{Me}_3\text{EtSb}[\text{Pd}(\text{dmit})_2]$ both show the Wilson ratio of approximately 1.6, which indicates moderate but appreciable correlations among fermions. Thus, the anomalous value, 0.63, for the present material must result from the doping and, in the manuscript, we argued this puzzling issue in terms of nearly decoupled spin and charge degrees of freedom. This open issue is awaiting to be tackled theoretically.

Comment 3-2:

My only criticism is the method chosen to display the non-Fermi liquid to Fermi liquid crossover via a contour plot, where the authors display the rate of change of the inter-planar resistivity. Why don't they mention, for example, the exponent of the planar resistivity in the non-Fermi liquid region? It seems to be linear in temperature. Even better, can they also display the exponent of the planar resistivity, as extracted from the derivative of $\log(\rho - \rho_0) = n \log(T)$? Yes I agree that this method yields n values that rely heavily on the actual value of ρ_0 . But there are ways of circumventing this. The value of n is directly correlated with the Fermi liquid and the non-Fermi liquid behavior(s).

Reply 3-2:

We agree that the value of n in the data fitting by the form of $\rho_{||} = \rho_0 + AT^n$, which characterizes the non-Fermi liquid nature, should be mentioned in the manuscript. Thus, we performed the fitting for the data of $\rho_{||}$ in Figs. 2b and 4b. In the revised manuscript, we added a figure (Fig. 3c), which shows the n values against pressure and referred to the behavior in the main text. As the reviewer suggested, a crossover from a non-Fermi liquid to a Fermi liquid is clear. The detail of the analysis is described in the new section "Power-law analysis of in-plane resistivity" with Supplementary Fig 3 in the Supplementary Information.

REVIEWERS' COMMENTS:

Reviewer #3 (Remarks to the Author):

The authors have addressed my point in terms of the evolution of the exponent in the resistivity as a function of the pressure.

Since the Heisenberg Hamiltonian with antiferromagnetic correlations in a triangular lattice is predicted/known to order (e.g. see Ref. 19 within the manuscript), these results are quite intriguing. Particularly, if one considers the fact that the authors use the same Hamiltonian to fit their spin susceptibility.

These results will be controversial among theorists, and should trigger a lot of activity. Hence, I will just have to recommend its publication.